



# Design optimization of offshore wind jacket piles by assessing support structure orientation relative to metocean conditions

Maciej M. Mroczek[1], Sanjay Raja Arwade[2,*], and Matthew A. Lackner[3,*]

[1]DEME Offshore, Scheldedijk 30, 2070 Zwijndrecht, Belgium
[2]Department of Civil and Environmental Engineering, University of Massachusetts - Amherst, 130 Natural Resources Road, Amherst, Massachussetts 01003, USA
[3]Department of Mechanical and Industrial Engineering, University of Massachusetts - Amherst, 160 Governors Drive, Amherst, Massachussetts 01003, USA
[*]These authors contributed equally to this work.

**Correspondence:** Maciej M. Mroczek (Mroczek.Maciej@deme-group.com)

**Abstract.** The orientation of a 3-legged offshore wind jacket structure in 60 meter water depth, supporting the IEA 15MW reference turbine, has been assessed for optimizing the jacket pile design. A reference site off the coast of Massachusetts was considered, including site specific metocean conditions and realistically plausible geotechnical conditions. Soil-structure interaction was modelled using three dimensional FE ground-structure simulations to obtain equivalent mudline springs which

were subsequently used in nonlinear elastic simulations, considering aerodynamic and hydrodynamic loading of extreme sea states in the time domain. Jacket pile loads were found to be sensitive to the maximum 50 year wave direction, as opposed to the wind direction, indicating that the jacket orientation should be considered relative to the dominant wave direction. The results further demonstrated that the jacket orientation has a substantial impact on the overall jacket pile mass and maximum pile embedment depth, and therefore represents an important opportunity for project cost and risk reductions. Finally, this research

highlights the importance of detailed knowledge of the full global model behavior (both turbine and foundation) for capturing this optimization potential, particularly due to the influence of wind-wave misalignment on pile loads. Close collaboration between the turbine supplier and foundation designer, at the appropriate design stages, is essential.

## 1 Introduction

The offshore wind industry remains relatively young and in a maturing phase, especially in the United States, with the main

characteristics of offshore wind farms continuing to grow in scale with each passing year. A recent survey by Beiter et al. (2022) of key industry participants (developers, turbine suppliers, universities, etc.) makes clear that the industry is expecting this trend to continue into the foreseeable future, with expectations of larger wind farms utilizing larger turbines in deeper water depths.

The dominant offshore wind foundation concept is the monopile foundation (Musial et al., 2022). As the industry continues

to scale, however, other foundation concepts are proving more economical for certain site conditions. The 2022 Offshore Wind Market Report (Musial et al., 2022), released by the U.S. Department of Energy, reports a global market trend towards





increasing diversity of foundation concepts, with the jacket concept, the subject of this paper, being the preferred concept for 11.8 GW (13.5%) of announced future projects. Additionally, jacket substructures were selected and constructed for the first US offshore wind farm at Block Island.

Jacket concepts for offshore wind application are generally found to either consist of three legs or four legs. While each variant has its unique advantages and disadvantages, which are best assessed on a site-specific basis, several previous studies have demonstrated generic benefits of the 3-legged jacket over the 4-legged variant (Tran et al., 2022; Chew et al., 2014, 2013). Building upon these previous findings, the 3-legged jacket concept will be explored further in this paper.

There is continuous pressure on the offshore wind industry to reduce the overall cost of offshore wind projects. Optimizing 30 the design of offshore foundations is one way to realize such cost reductions. As most offshore sites are characterized by dominant wind and wave directions, the orientation of the jacket relative to these metocean conditions will influence the loads experienced by the foundation structure and, therefore, the final foundation cost. Several past studies have examined the influence of jacket orientation on the jacket structure design, including Tran et al. (2022, 2021); Wei et al. (2017, 2016); Chew et al. (2014, 2013). All of these studies found that the jacket design is sensitive to orientation relative to loading direction.

For 3-legged jackets, Tran et al. (2022) found that the jacket leg stresses were lowest when one leg pointed away from the oncoming wind and wave direction. Chew et al. (2014) found that ultimate limit state (ULS) and fatigue limit state (FLS) utilization ratios for the 3-legged jacket varied per joint type. Generally, the ULS and FLS utilization ratios in the legs and X-braces were optimized when one leg pointed into the oncoming wind-wave direction, while the Y-/T- and K-braces generally performed better when one leg pointed away from the oncoming wind-wave direction.

While the previously cited studies focused on the jacket structure design, there was no particular attention paid to the impact of jacket orientation on the jacket pile design. Furthermore, the past work has been performed considering either 3MW or 5MW reference turbines, which are substantially smaller than current industry sizes. This paper will expand on the previous work and fill in these gaps by focusing on the influence of jacket orientation on the pile design of a 3-legged jacket in 60 meter water depth supporting the IEA 15 MW reference turbine.

The purpose of this research is to identify the optimal jacket orientation for pile design and draw conclusions that can be applied in the industry for realizing design optimizations and overall project cost savings. A reference site off the coast of Massachusetts has been considered for this work. The remainder of the paper is organized as follows: Section 2 provides an overview of the site conditions; Section 3 lays out the global modeling approach, including the turbine characteristics, jacket structure design, and soil-structure interaction. Details regarding the time domain dynamic simulations are provided in Section 50 4; Finally, results and conclusions are presented in Sections 5 and 6, respectively.

## 2 Site Conditions

A generic reference site off the coast of Massachusetts is considered for this paper. The North East coast of the United States is generally well suited for offshore wind applications, and has several offshore wind projects under active development in the area. The following subsections provide more details on the metocean and geotechnical conditions of the reference site.





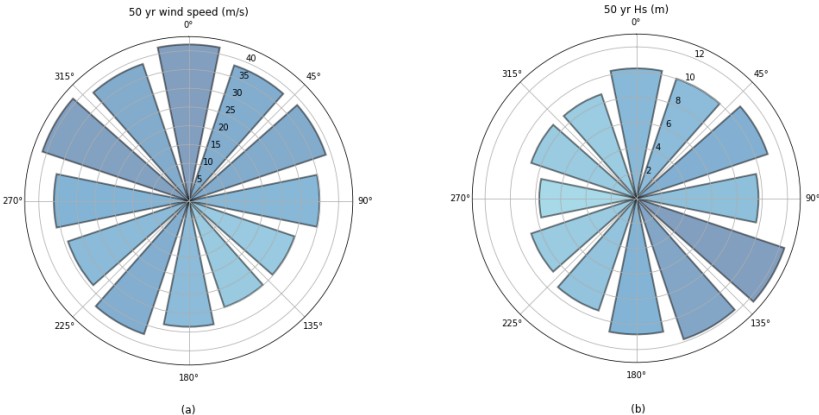

**Figure 1.** 50 year wind speed (a) and significant wave height (b) per directional bin

## 2.1 Metocean conditions

To derive the metocean conditions, data from a NOAA buoy located 54 nautical miles southeast of Nantucket (station 44008) was processed. This buoy provides historical wind and wave data from 1982 to date. This particular buoy was selected due to its relative proximity to existing offshore wind farm leasing areas.

The buoy data was processed using the method elaborated by Gringorten (1963), as described by Rohatgi et al. (2013) and Palutikof et al. (1999). 50 year significant wave height and wind speed values per direction, in 30 degree increments, were determined by first identifying the yearly maxima per direction from the available data record. Values from years with less than 90% coverage of wind speed and wind direction were disregarded, resulting in 15 years of data coverage. Wave data coverage was lower than wind data, therefore a lower threshold of 70% coverage was used for wave height and wave direction, in order to have 10 years of applicable data.

These values were then ranked according to the Gringorten method, per 30 ° directional bin, and a linear regression was performed. The wind speed at buoy level was converted to hub height using the wind shear power law profile and an alpha value of 0.11. The 50 year maximum values per directional bin are presented in the rose diagrams in Figure 1.

The NOAA buoy does not measure current data. Therefore, a very conservative current speed of 2 m/s aligned with wave direction was assumed. Guidance provided in the DNV-OS-J101 standard was used to determine the associated wave period ($T$) and maximum wave height ($H_{max}$) for a given significant wave height ($H_S$):

$$11.1\sqrt{H_S/g} \le T \le 14.3\sqrt{H_S/g} \text{ (average value used)}$$

$$H_{max} = 1.86H_S$$

In order to determine potential wind-wave misalignment during extreme metocean conditions, the NOAA buoy data was filtered for occurrences of wind speeds in excess of 25 m/s combined with significant wave heights above 8 m. There were 48 occurrences of this combination available in the NOAA buoy data. The number of occurrences per wind-wave misalignment angle is presented in Figure 2.

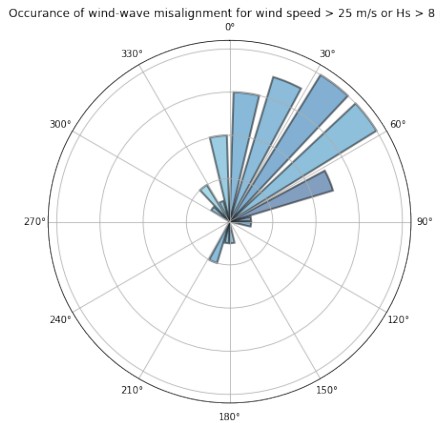

**Figure 2.** Occurrence of wind-wave misalignment during extreme metocean conditions

**Table 1.** Metocean Cases

|  | Wind Speed (m/s) | Wind Direction (degrees) | Sign. Wave Height (m) | Wave Direction (degrees) | Wind-Wave Misalign. (degrees) |
|---|---|---|---|---|---|
| Metocean Case 1 | 41.7 | omni- | 12.4 | omni- | 0 |
| Metocean Case 2 | 41.7 | 0 | 10.0 | 30 | 30 |
| Metocean Case 3 | 41.7 | 0 | 11.0 | 60 | 60 |
| Metocean Case 4 | 34.9 | 90 | 12.4 | 120 | 30 |
| Metocean Case 5 | 38.6 | 60 | 12.4 | 120 | 60 |
| Metocean Case 6 | 41.7 | 0 | 12.4 | 120 | 120 |

This particular site typically experiences between 15 - 60 degree wind-wave misalignment during extreme sea states, however, larger wind-wave misalignments also occurred. Six potential metocean cases, shown in Table 1, are derived from this post-processed data and further considered for this reference site.

Case 1 considers the omnidirectional maximum 50-year wind and wave conditions without wind-wave misalignment. Cases 2 and 3 consider the highest wind speed with 30° and 60° wind-wave misalignment and associated wave conditions. Cases 4 and 5 consider the worst-case 50-year waves with 30° and 60° wind-wave misalignment and associated wind speeds.

Finally, Case 6 considers the maximum 50-year wind and wave conditions according to their actual direction (as opposed to the omnidirectional approach of Case 1), which results in a wind-wave misalignment of 120°. Case 6 is intended to represent misalignment angles greater than 60° which, though rare, have occurred within the forty year history of the buoy data.



**Table 2.** Representative Soil Profile

| Layer Extent | Unit Description | Friction Angle (degrees) | Effective Unit Weight ($kN/m^3$) |
| --- | --- | --- | --- |
| 0 - 10 m | Medium-Dense Sand | 33° | 8.8 |
| 10 m + | Dense Sand | 37° | 9.8 |

## 2.2 Geotechnical conditions

BOEM Office of Renewable Energy commissioned Fugro to prepare a report with a geological and geotechnical overview of the Atlantic and Gulf of Mexico Outer Continental Shelf regions (Trandafir et al., 2022). This report includes idealized and generic soil profiles for the New England Shelf Region. Soil Profile 1 for this region is characterized as a Holocene Marine and / or Transgressive sand layer, with a thickness of approximately 10 m, overlaying a thick glacial drift sand. The report also provides a range of inferred geotechnical parameters for these two soil layers. A representative soil profile was selected considering the guidance of this report, as shown in Table 2.

## 3 Global Model

In order to investigate the impact of the support structure orientation on the jacket pile design, dynamic simulations were run in the OpenFAST software package (NREL, a). OpenFAST is a software tool used for nonlinear elastic simulations of wind turbines subjected to user defined aerodynamic and hydrodynamic loading conditions in the time domain. OpenFAST also includes capabilities for dynamic simulations of the turbine's control and electrical systems.

The currently available OpenFAST version does not allow for the implementation of wave stretching beyond the still water surface elevation. This limitation would lead to unrealistic results as the forces on the structure caused by wave particle velocities above the still water level would be omitted. Therefore, the OpenFAST source code was modified to allow for wave stretching, via the extrapolation method, by making use of the new SeaState module (NREL, b).

For this work, the structural dynamics of the global model were computed using OpenFAST's ElastoDyn and SubDyn modules, while aerodynamic loads were computed using the AeroDyn v15 module. The sea state and hydrodynamic loads were calculated using the SeaState and HydroDyn modules of OpenFAST.

### 3.1 Turbine Characteristics

The IEA 15MW reference turbine is used for this analysis as it is representative of the currently available commercial offshore wind turbines (Gaertner et al., 2020). Key parameters of the IEA 15MW reference turbine are provided in Table 3.

The IEA 15MW reference turbine includes a standard tower design, however, this tower length was shortened in order to allow for a higher interface level between the jacket and tower. The new interface level was set to 30 m above still water level, in order to locate the bottom of the foundation transition piece above the maximum wave crest elevation (air gap requirement).





**Table 3.** IEA 15MW key parameters (Gaertner et al., 2020)

| Parameter | Units | Value |
|---|---|---|
| Power Rating | MW | 15 |
| Control | - | Variable speed Collective Pitch |
| Min Rotor Speed | rpm | 5.0 |
| Max Rotor Speed | rpm | 7.56 |
| Rotor Diameter | m | 240 |
| Hub height | m | 150 |
| Blade mass | t | 65 |
| RNA mass | t | 1017 |

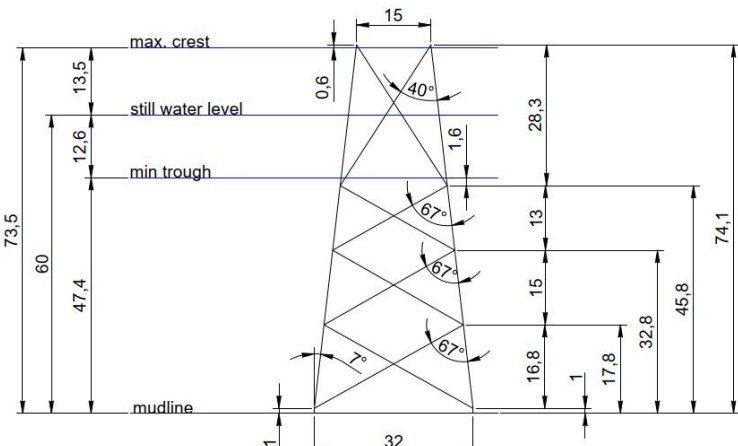

**Figure 3.** Main jacket dimensions

## 3.2 Jacket Design

The main dimensions, shown in Figure 3, of the 3-legged jacket structure were chosen to be representative for 60 meter water depth. A transition piece with height of 15.9 m and mass of 450 metric ton was modeled as a rigid and locked connection between the top of the jacket legs and bottom of the turbine tower.

The implementation of wave stretching in the modified OpenFAST source code requires that surface-piercing members do not become submerged throughout the simulation, while fully submerged members must remain fully submerged. This had an influence on the jacket design, requiring that the upper most X-bay frame have a larger height (28.3 m) than would be structurally optimal. This feature of the jacket design should have minimal effect on the pile-top loads.



**Table 4.** Jacket member dimensions

| Description | Diameter (m) | Thickness (mm) |
|---|---|---|
| Upper-most Brace Members | 1.1 | 70 |
| Remaining Brace Members | 1.0 | 20 |
| Upper-most Brace Frame Legs | 1.2 | 52 |
| 2nd Brace Frame Legs | 1.4 | 45 |
| 3rd Brace Frame Legs | 1.5 | 48 |
| Lowest Brace Frame Legs | 2.0 | 50 |
| Legs at mudline | 3.0 | 50 |

**Table 5.** Estimated pile top loads

| Description | Tension | Compresson | Shear | Moment | Torsion |
|---|---|---|---|---|---|
| | 37.5 MN | 64.0 MN | 13.8 MN | 19.7 MN-m | 4.5 MN-m |

Although the jacket structure is not the focus of this research, a sensible design of the jacket members is necessary for determining realistic jacket pile loads. Therefore, the jacket members were sized according to the Norsok N-004 standard using preliminary ULS member forces generated in OpenFAST using simplified assumptions. The simulations were run iteratively until conservative utilization ratios were reached. The resulting jacket member sizes are provided in Table 4. These member sizes are generally in accordance with past experience, however, the upper-most brace frame elements have a large thickness due to the abnormally large bay height. The overall jacket mass is 1471 MT, excluding transition piece and jacket piles. The jacket structure was incorporated into the global model using a Craig-Bampton reduction with six retained internal modes.

The Morison coefficients for the jacket members were set according to the general recommendations in API RP 2A-WSD for a rough, unshielded circular cylinder: $C_d = 1.05$ and $C_m = 1.2$. Marine growth was also assumed to be present on the jacket structure from +13 m relative still water level (SWL) to -40m SWL with 50mm thickness and density of 1325 kg/m$^3$. The jacket legs are modeled as flooded members.

## 3.3 Soil Structure Interaction

An initial estimate of the jacket pile design was used to assess the soil structure interaction of the global model. The outer diameter and thickness of the pile were assumed to be 3.0 m and 50 mm, respectively, with a pile embedment depth below seabed of 65 m.

A reduced number of simulations in OpenFAST were then run with the jacket clamped at the seabed. These simulations considered the maximum 50 year wind and wave conditions with 0° and 30° wind-wave misalignment, a yaw error of -120°, a jacket orientation of 30°, and six wind and wave seeds. The resulting maximum pile top loads in tension, compression, shear, bending, and torsion from these simulations are shown in Table 5.





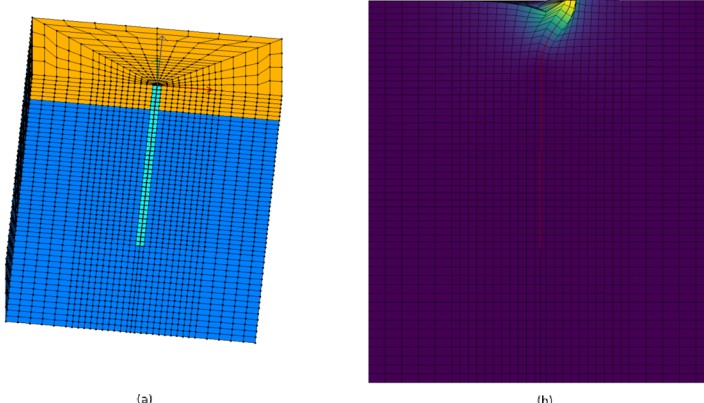

**Figure 4.** OpenSeesPL half-space mesh (a) and deformed mesh under shear loading (b)

OpenSeesPL (Lu et al., 2011), software for 3D finite element modeling of ground-structure response, was used to determine the mudline stiffness of the pile in the reference soil profile using these loads with static push over analyses. Figure 4(a) shows the half-space mesh used for the FE modeling. A full-space mesh was used for assessing torsional stiffness, whereas the half-space mesh was used for the other stiffness components.

Static push over analyses were run to determine the displacement of the pile head for all of the estimated pile top loads. Figure 4(b), for example, shows the deformed mesh under shear loading, which was used to determine the lateral stiffness components. The FE modeling results were used to determine the stiffness matrix of the jacket piles at mudline. Table 6 shows the values derived from these analyses, which were used further in the OpenFAST simulations to account for the soil structure interaction. A mass matrix was also derived to account for the mass of the jacket piles. The mass matrix assumed that the mass of each jacket pile is concentrated equally at both of the member ends.

Finally, free-decay testing of the global model, including the soil structure interaction, was simulated in OpenFAST. This analysis showed the 1st natural frequency of the global model to be 0.211 and 0.213 in side-side and fore-aft directions, respectively. These values fall outside the 1P and 3P frequency ranges of the IEA 15MW turbine and, therefore, the jacket frequency is according to expectations and valid for further simulations.

## 4 Simulation Setup

### 4.1 Design Load Cases

Design load cases (DLCs) 6.1 and 6.2 are considered in the dynamic OpenFAST simulations, as specified in the IEC 61400-3 standard. These DLCs simulate a parked and feathered turbine under extreme 50 year environmental conditions and varying yaw errors (Table 7). These DLCs were selected as the most likely governing load cases for the jacket pile embedment depth





**Table 6.** Stiffness matrix accounting for the soil-structure interaction

|  | X-disp | Y-disp | Z-disp | X-rot | Y-rot | Z-rot |
|---|---|---|---|---|---|---|
| X-disp | 1.18 (GN/m) | 0 | 0 | 0 | -5.12 (GN/rad) | 0 |
| Y-disp | - | 1.18 (GN/m) | 0 | 5.12 (GN/rad) | 0 | 0 |
| Z-disp | - | - | 18.8 (GN/m) | 0 | 0 | 0 |
| X-rot | - | - | - | 36.4 (GN-m/rad) | 0 | 0 |
| Y-rot | - | - | - | - | 36.4 (GN-m/rad) | 0 |
| Z-rot | - | - | - | - | - | 10.4 (GN-m/rad) |

**Table 7.** Description of considered design load cases

| DLC | Metocean Conditions | Yaw Error | Load Factor |
|---|---|---|---|
| 6.1 | 50 yr wind & waves | +8°, 0°, -8° | 1.35 |
| 6.2 | 50 yr wind & waves | 30° increments | 1.1 |

because the load assessment of the IEA 15MW reference turbine (Gaertner et al., 2020) shows that the parked load cases result in the highest overall tower base moment.

Previous studies (Niranjan and Ramisetti, 2022; Morató et al., 2017) have reported unrealistic behavior in aeroelastic simulations of wind turbine blades under certain yaw errors. The cause of this phenomenon is understood to be related to an overprediction of blade vibrations in deep stall (Skrzypiński and Gaunaa, 2015). Therefore, yaw errors which lead to unrealistic results, $\pm30°$ and $\pm60°$ yaw errors for this particular configuration, have been neglected.

Each individual simulation had a duration of 600 seconds. In order to avoid longer simulation durations, according to IEC 61400-3, a constrained wave with guaranteed peak-to-trough crest height was incorporated into the time series at 400 seconds using NREL's SeaState module. The initial 200 seconds of each simulation were discarded to eliminate transient start-up effects.

The inflow wind time series was generated using the TurbSim software (NREL). The IEC Kaimal model was considered with a turbulence intensity of 11% and a power law wind shear profile. The wave time series is generated directly within OpenFAST using the JONSWAP spectrum (Hasselmann et al., 1973). Six different random seeds were used for generating the wind and wave time series.



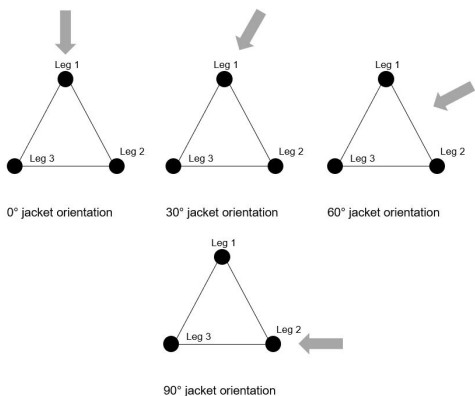

**Figure 5.** Overview of the jacket orientations considered ($30°$ and $90°$ orientations are symmetrical)

## 4.2 Jacket Orientations

As directional metocean data is typically provided in $30°$ increments, the jacket orientation is also considered in these increments. Due to symmetry of the three-legged jacket structure, a full range of possible jacket orientations can be considered by rotating the jacket structure from $0°$ to $120°$. Furthermore, the $30°$ and $90°$ orientations are also symmetrical. Therefore, the

jacket orientations considered are $0°$, $30°$, and $60°$, which covers the full range of jacket orientations in $30°$ increments. It should be noted that the global model does not include a boat landing, which would not have allowed for such symmetry.

OpenFAST provides a method for directly setting the wind and wave propagation directions, whereas rotating the substructure is more cumbersome. For this reason, the jacket orientations were simulated in OpenFAST by rotating the governing wind and wave directions relative to the jacket structure. Figure 5 provides an illustrated view of the jacket orientations considered

in this final project.

## 5 Results

In total, 1080 simulations were performed in OpenFAST to evaluate the influence of jacket orientation on the pile load and design. The simulations considered three jacket orientations, six wind and wave seeds, six metocean cases and ten nacelle yaw errors.

Tables 8 and 9 show the resulting maximum pile top tension and compression forces, respectively, for the three jacket orientations. These are maximum values averaged over the six wind and wave seeds; the appropriate load factor per DLC is also included. The orientations are provided relative to wave direction, the reason for this is further explained in the following subsection. Metocean case 2 did not include a $0°$ orientation relative to wave direction, as cases 2 and 3 were simulated considering the jacket orientation relative to the wind direction.



**Table 8.** Maximum seed-averaged pile-top tension (orientations relative to wave direction)

|                  | 0° orientation | 30° orientation | 60° orientation |
|------------------|----------------|-----------------|-----------------|
| Metocean Case 1  | 39.2 MN        | 31.8 MN         | 18.1 MN         |
| Metocean Case 2  | na             | 20.9 MN         | 13.8 MN         |
| Metocean Case 3  | 30.2 MN        | 25.6 MN         | 15.7 MN         |
| Metocean Case 4  | 38.4 MN        | 30.9 MN         | **19.0 MN**     |
| Metocean Case 5  | 38.3 MN        | 29.8 MN         | 16.8 MN         |
| Metocean Case 6  | **40.9 MN**    | **33.5 MN**     | 17.0 MN         |
| **Maximum**      | 40.9 MN        | 33.5 MN         | 19.0 MN         |

**Table 9.** Maximum seed-averaged pile-top compression (orientations relative to wave direction)

|                  | 0° orientation | 30° orientation | 60° orientation |
|------------------|----------------|-----------------|-----------------|
| Metocean Case 1  | **50.6 MN**    | 61.9 MN         | 66.2 MN         |
| Metocean Case 2  | na             | 50.9 MN         | 51.8 MN         |
| Metocean Case 3  | 44.0 MN        | 53.8 MN         | 61.3 MN         |
| Metocean Case 4  | 49.0 MN        | **63.9 MN**     | 67.5 MN         |
| Metocean Case 5  | 47.1 MN        | **63.9 MN**     | 69.0 MN         |
| Metocean Case 6  | 49.0 MN        | 62.5 MN         | **70.9 MN**     |
| **Maximum**      | 50.6 MN        | 63.9 MN         | 70.9 MN         |

## 5.1 Optimal Jacket Orientation

The reference site conditions show that the dominant wind and wave directions are approaching from different directions. Therefore, when seeking to optimize the jacket pile design, it is important to first understand whether the jacket orientation should be set relative to the wind or wave direction. Comparing the metocean cases that considered highest wind speed combined with lower waves (cases 2 and 3) with the cases that considered largest waves with lower wind (cases 4 and 5), the cases with largest waves resulted in the highest tension and compression loads (Table 10). This demonstrates that the wave forces are dominant for the pile axial loads under the considered design load cases.

Furthermore, when considering which orientation relative to wind leads to minimum and maximum pile axial loads, no clear conclusion can be drawn (Table 11). However, the results are consistent when considering the jacket orientation relative to the wave direction (Table 12). This further demonstrates that the wave forces are dominant for the pile axial loads and should be used for the jacket orientation.

Maximum tension and minimum compression occurs when the jacket is orientated with one leg facing into the oncoming wave direction (0° orientation). The results are reversed when one leg is opposite the oncoming wave direction (60° orientation). However, it is possible that these results may vary for sites with different metocean conditions than those considered in this study.





**Table 10.** Maximum pile axial loads for metocean cases with highest wind speed vs largest waves

|  | Description | Max Tension | Max Compression |
| --- | --- | --- | --- |
| Metocean Cases 1 & 2 | Highest wind speed | 30.2 MN | 61.3 MN |
| Metocean Cases 3 & 4 | Largest waves | 38.4 MN | 69.0 MN |
|  | **Percent Difference** | 23.9% | 11.8% |

**Table 11.** Jacket orientations (relative to wind direction) that lead to max and min pile axial loads

|  | Max Tension | Min Tension | Max Compression | Min Compression |
| --- | --- | --- | --- | --- |
| Metocean Case 2 | 0° | 30° | 30° | 60° |
| Metocean Case 3 | 60° | 0° | 0° | 60° |

## 5.2 Impact on Jacket Pile Design

The resulting axial loads were then used to calculate the required pile length using the API method, in accordance with the DNV-RP-C212 standard, considering the reference site soil profile. Pile mass was also estimated assuming a constant wall thickness of 50mm for the piles. The results are provided in Table 13.

These results demonstrate that orientating the jacket into the oncoming wave direction reduces the maximum pile embedment depth by -9.0% and -21.1%, compared to the other two orientations, for these reference site conditions. Reducing pile embedment depth may help mitigate projects risks, such as the risk of pile refusal during driving. It may also be beneficial in case of particularly challenging soil conditions (i.e., rock layers) below a certain depth. Furthermore, this orientation resulted in an estimated pile mass savings of 9 - 10% compared to the other orientations.

## 5.3 Wind-wave misalignment

Metocean case 1 considered an omnidirectional approach whereby the maximum 50 year wind and wave conditions are applied without misalignment. The remaining cases considered varying amounts of wind-wave misalignment, as shown in Table 1. To investigate the impact of wind-wave misalignment on the jacket pile design, the differences for pile tension and compression forces are presented in Tables 14 and 15, respectively.

The maximum pile loads were underestimated for the majority of jacket orientations when wind-wave misalignment was not considered. The largest difference was observed for the 60° orientation in compression. Metocean case 6, with a wind-wave misalignment of 120°, resulted in the highest compression force in that orientation. To better understand the potential cause of this difference, the maximum compression loads (including load factor) for this orientation and metocean case are shown per yaw error in Figure 6.



**Table 12.** Jacket orientations (relative to wave direction) that lead to max and min pile axial loads

|  | Max Tension | Min Tension | Max Compression | Min Compression |
|---|---|---|---|---|
| Metocean Case 1 | 0° | 60° | 60° | 0° |
| Metocean Case 3 | 0° | 60° | 60° | 0° |
| Metocean Case 4 | 0° | 60° | 60° | 0° |
| Metocean Case 5 | 0° | 60° | 60° | 0° |
| Metocean Case 6 | 0° | 60° | 60° | 0° |

**Table 13.** Pile length and mass for the calculated maximum axial loads

|  | Leg | Max Loads | Pile Embedment | Est. Pile Mass |
|---|---|---|---|---|
| **0° orientation** | 1 | 48.6 MN (C) / 39.2 MN (T) | 63.0 m | 229 MT |
|  | 2 | 47.4 MN (C) / 41.0 MN (T) | 65.4 m | 238 MT |
|  | 3 | 49.8 MN (C) / 1.7 MN (T) | 47.9 m | 174 MT |
|  |  |  | **Total** | **641 MT** |
| **30° orientation** | 1 | 64.6 MN (C) / 31.8 MN (T) | 71.9 m | 262 MT |
|  | 2 | 34.4 MN (C) / 33.6 MN (T) | 55.6 m | 202 MT |
|  | 3 | 61.9 MN (C) / 2.0 MN (T) | 67.5 m | 246 MT |
|  |  |  | **Total** | **710 MT** |
| **60° orientation** | 1 | 71.4 MN (C) / 18.0 MN (T) | 82.9 m | 302 MT |
|  | 2 | 33.1 MN (C) / 19.3 MN (T) | 36.6 m | 133 MT |
|  | 3 | 66.2 MN (C) / 15.5 MN (T) | 74.5 m | 270 MT |
|  |  |  | **Total** | **705 MT** |

The maximum compression load occurs when the turbine encounters an 8° yaw error relative to the oncoming wind direction.
To better understand the behavior of the turbine at this yaw error, wind only simulations of the global model were run using



**Table 14.** Maximum seed-averaged pile tension with and without wind-wave misalignment

|  | 0° orientation | 30° orientation | 60° orientation |
| --- | --- | --- | --- |
| Wind & waves aligned | 39.2 MN | 31.8 MN | 18.1 MN |
| Wind & waves misaligned | **40.9 MN** | **33.5 MN** | **19.0 MN** |
| **Percent difference** | 4.2% | 5.2% | 4.9% |

**Table 15.** Maximum seed-averaged pile compression with and without wind-wave misalignment

|  | 0° orientation | 30° orientation | 60° orientation |
| --- | --- | --- | --- |
| Wind & waves aligned | **50.6 MN** | 61.9 MN | 66.2 MN |
| Wind & waves misaligned | 49.0 MN | **63.9 MN** | **70.9 MN** |
| **Percent difference** | 3.2% | 3.2% | 6.9% |

the maximum 50 year wind speed. The maximum tower bending moment, per 30° directional bin, with an 8° yaw error was determined from these simulations and shown in Figure 7.

The maximum tower bending moment for this yaw error occurs in the side-to-side direction, which explains why the maximum forces for this yaw error are underestimated when excluding wind-wave misalignment. A comprehensive understanding of the turbine behavior is essential for determining the wind-wave misalignments that may result in maximum pile loads.

## 6 Conclusions

This research has focused on examining the influence of jacket orientation, relative to metocean conditions, on the jacket pile design. First, it was observed that jacket orientation should be assessed relative to the wave direction, as opposed to wind direction, as the wave forces are dominant for pile loads. These findings are valid for the reference site conditions used in the study. The wind direction may become more dominant when considering different conditions, such as different wind turbine types or sites with higher extreme wind speeds or lower extreme wave heights.

Orientating the jacket such that one leg faces into the oncoming wave direction was shown to reduce the maximum jacket pile embedment depth and total pile mass. Designing for this orientation could not only reduce project costs in terms of reduced steel tonnage, but could further reduce project risks such as pile refusal or potential need for drilling into deep rocky soil layers. The impact of orientation on the jacket structure design, as opposed to the jacket piles, should also be considered. This was not the focus of this paper and has been investigated in detail in previous studies by others.

Finally, it was found that the omnidirectional metocean case, which considered highest wind and wave conditions but did not account for wind-wave misalignment, did not consistently result in the highest axial pile loads. The loads from this case were underestimated by up to 6.9% compared to cases including wind-wave misalignment. The cause of this difference could be traced back to the behavior of the turbine subjected to wind loading under various yaw errors. Caution must be exercised

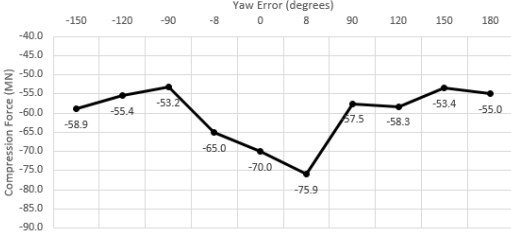

**Figure 6.** Max compression force per yaw error for 60° jacket orientation and metocean case 6

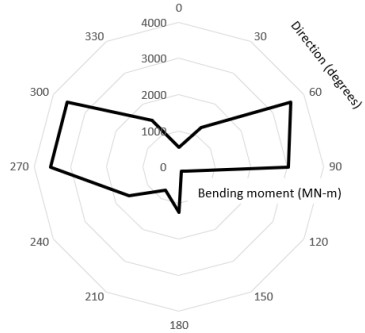

**Figure 7.** Polar diagram of tower bottom bending moment with 8° yaw error (0° oncoming wind direction)

when considering whether or not to include wind-wave misalignment in a reduced load case table, for example during design stages prior to detailed design.

This study demonstrates that selecting an optimal jacket orientation relative to the site specific metocean conditions can provide an important design optimization. However, this requires close collaboration between the turbine supplier and foundation designer. As the pile loads are influenced by both the turbine and foundation behavior, this optimization cannot be studied individual and requires detailed knowledge of the full global model.

*Author contributions.* The study approach and objectives were jointly developed by the authors. MM performed the calculations and simulations. Conclusions, based on analysis of the results, were jointly developed. MM prepared the manuscript with contributions from all co-authors.

*Competing interests.* The contact author has declared that none of the authors has any competing interests.



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
