# Peer review of "Design optimization of offshore wind jacket piles by assessing support structure orientation relative to metocean conditions"

_Wind Energy Science, 2023_

## Author Response (AR1)

Dear Thanh-Tuan Tran and Referee #2,

Thank you both for your review of our work and for your constructive feedback. This pdf provides our clarifications to your comments and provides an exhaustive overview of all changes introduced in this latest revision of our article.

**Reply to comments received on 02 Feb 2023 from Thanh-Tuan Tran**

*#1: Page 7: Where are the dimensions of the jackets (jacket properties) from? Were these derived from the existing jacket substructure of the IEA 15MW OWT? Please briefly summazing the jacket preliminary design.*

**Reply:** The jacket dimensions are not derived from a pre-existing jacket substructure design. The third paragraph in section 3.2 provides a summary of the jacket design, which we hope is sufficiently clear: "the jacket members were sized according to the Norsok N-004 standard using preliminary ULS member forces generated in OpenFAST using simplified assumptions. The simulations were run iteratively until conservative utilization ratios were reached. The resulting jacket member sizes are provided in Table 4. These member sizes are generally in accordance with past experience, however, the upper-most brace frame elements have a large thickness due to the abnormally large bay height."

*#2: Do the dominant frequencies of the whole system check in the preliminary design? And what are the frequency ranges for 1P and 3P from? Source is missing.*

**Reply:** The 1P and 3P frequency ranges of the IEA 15MW turbine are not explicitly stated in the IEA report (Gaertner et al., 2020), but can be determined from the min and max rotor speeds (shown in Table 4 of our article and sourced from the IEA report). The 1P and 3P frequency ranges are 0.083 - 0.126 Hz and 0.249 – 0.378 Hz, respectively. These forcing frequency ranges do not coincide with the first natural frequency of the global model: 0.211 and 0.213 in side-side and fore-aft directions, respectively (as reported in section 3.3).

**Revision to article:** changes to the last paragraph in section 3.3 are highlighted in yellow:

> Finally, free-decay testing of the global model, including the soil structure interaction, was simulated in OpenFAST. This analysis showed the 1st natural frequency of the global model to be 0.211 Hz and 0.213 Hz in side-side and fore-aft directions, respectively. These values fall outside the 1P and 3P frequency ranges (0.083 - 0.126 Hz and 0.249-0.378 Hz, respectively) of the IEA 15MW turbine and, therefore, the jacket frequency is according to expectations and valid for further simulations.

*#3: The coordinate system of the model must be illustrated. Currently it is very hard to work out the applied moment and force loads. Where are the loads exactly applied.*

**Reply:** Figure 3 is replaced with an updated image showing the coordinate system (a copy of the figure is also provided below for your convenience). Loads reported in the article are pile top loads of the jacket

piles. Internal loads of the jacket structure are not reported. We also try to make the distinction between the jacket structure and the jacket piles clearer in the update to Figure 3.

In the OpenFAST simulations, the environmental loads are applied to the global model and the resulting pile top loads are calculated by the software. In the OpenSeesPL simulations, ref. Figure 4, the pile top-loads are applied at the pile top (i.e., mudline).

**Revision to article:** Figure 3 is replaced with the follow image and changes to the associated caption are highlighted in yellow:

[Figure]

**Figure 3.** Main jacket dimensions (a) and coordinate system (b)

Changes to the 2nd paragraph of section 3.3 are highlighted in yellow:

> A reduced number of simulations in OpenFAST were then run with the jacket clamped at the seabed. These simulations considered the maximum 50 year wind and wave conditions with 0° and 30° wind-wave misalignment, a yaw error of -120°, a jacket orientation of 30°, and six wind and wave seeds. The resulting maximum pile top loads in tension, compression, shear, bending, and torsion from these simulations are shown in Table 5. The coordinate system corresponding to the pile top loads reported throughout this work is shown in Figure 3(b); the reported shear and moment loads are the resultant loads of their respective x- and y- components.

*#4: The results from parametric studies (Tables 8-9) should be graphical reported. The polar diagram, which shows how the response polarized under the metocean condition, is recommended*

**Reply:** The results previously reported in Tables 8 and 9 are now represented graphically in the update to the article (reproduced below for your convenience). Please note that these results cannot be shown in a polar diagram, since they are axial compression and tension loads acting along the same axis (zaxis). Hopefully the update to Figure 3 (particularly the coordinate system) helps to clarify any misunderstandings about the direction of the loading.

**Revision to article:** Table 8 is replaced with the following figure and associated caption:

[Figure]

**Figure 6.** Maximum seed-averaged pile-top tension (orientations relative to wave direction)

Table 9 is replaced with the following figure and associated caption:

[Figure]

**Figure 7.** Maximum seed-averaged pile-top compression (orientations relative to wave direction)

Changes to the 2nd paragraph of section 5 are highlighted in yellow:

> Figures 6 and 7 show the resulting maximum pile top tension and compression forces, respectively, for the three jacket orientations. These are maximum values averaged over the six wind and wave seeds; the appropriate load factor per DLC is also included. The orientations are provided relative to wave direction, the reason for this is further explained in the following subsection. Metocean case 2 did not include a 0° orientation relative to wave direction, as cases 2 and 3 were simulated considering the jacket orientation relative to the wind direction.

Figures 8 and 9 and Tables 8 – 13 are renumbered (previously figures 6 and 7 and Tables 10 - 15 in the preprint) due to these two tables being replaced by figures.

*#5: The use of 15MW Wind turbine with the reference site condition in this paper does not reflect for substructures with different topological configuratons (such as Pratt, Warren brace systems). The author could mention that what should be considered with others?*

**Reply:** Correct, X-bracing was considered due to the prevalent use of this bracing type in the offshore wind industry. Caution should be used when attempting to extrapolate the results of this study to different global model configurations, including bracing types but also other aspects (e.g., different turbine models, water depths, soil conditions, etc.). The optimal jacket orientation used in practice should be determined based on an assessment of the actual project specific information. Nevertheless, we believe our research has demonstrated that pursuing such assessments on a project-specific basis could provide meaningful project cost and risk reductions.

**Revision to article:** In order to make this point clearer in the text, the first two paragraphs of section 6 (conclusion) are revised. Changes to these paragraphs are highlighted in yellow:

> ## 6 Conclusions
>
> This research has focused on examining the influence of jacket orientation, relative to metocean conditions, on the jacket pile design. First, it was observed that jacket orientation should be assessed relative to the wave direction, as opposed to wind direction, as the wave forces are dominant for pile loads. Secondly, orientating the jacket such that one leg faces into the oncoming wave direction was shown to reduce the maximum jacket pile embedment depth and total pile mass. Designing for this orientation could not only reduce project costs in terms of reduced steel tonnage, but could further reduce project risks such as pile refusal or potential need for drilling into deep rocky soil layers.
>
> These particular results are valid for the reference site conditions, global model, and calculation methods used in the study. The optimal jacket orientation used in practice should be determined based on an assessment of the actual project specific information. Nevertheless, the results demonstrate that pursuing such assessments, on a project-specific basis, could provide meaningful project cost and risk reductions. The impact of orientation on the jacket structure design, as opposed to the jacket piles, should also be considered. This was not the focus of this paper and has been investigated in detail in previous studies by others.

**Reply to comments received on 07 Feb 2023 from Referee #2**

*#1: The derivation of the 50-yr omni directional extremes is overly simplified, and overestimated in terms of magnitude. The directional data should be combined and the extreme value analysis done over the entire history of wind/waves to obtain the omni-directional magnitudes, which are typically smaller than the worst case direction, as are used in this manuscript. The careful selection of directional bins and omni-directional data have a strong impact on environmental magnitudes, structural loads, and overall reliability, as examined in : Forristall, George Z. "On the use of directional wave criteria." Journal of waterway, port, coastal, and ocean engineering 130.5 (2004): 272-275. and Feld, Graham, Philip Jonathan, and David Randell. "On the estimation and application of directional design criteria." International Conference on Offshore Mechanics and Arctic Engineering. Vol. 58851. American Society of Mechanical Engineers, 2019.*

**Reply:** Thank you for this well spotted comment. The maxima of the directional extreme values, not the omni-directional extreme values, are intentionally used for metocean case 1. This is done in order to isolate the influence of wind-wave misalignment, as reported in section 5.3. With this objective in mind, it was necessary to use equal magnitudes for metocean cases 1 and 6, while only varying the misalignment. Nevertheless, as you rightfully point out, the maxima of the directional extreme values are not the same as the omni-directional extreme values. This inaccuracy in the wording has been corrected.

**Revision to article:** Changes to the 2nd to last paragraph of section 2.1 are highlighted in yellow:

> Case 1 considers the ==maxima of the directional== 50-year wind and wave ==values== without wind-wave misalignment ==by intentionally setting both the wind and wave directions to 0°==. Cases 2 and 3 consider the highest wind speed with 30° and 60° wind-wave misalignment and associated wave conditions. Cases 4 and 5 consider the worst-case 50-year waves with 30° and 60° wind-wave misalignment and associated wind speeds.

Deletion in the last paragraph in section 2.1 is shown below in red:

> Finally, Case 6 considers the maximum 50-year wind and wave conditions according to their actual direction (as opposed to  Case 1), which results in a wind-wave misalignment of 120°. Case 6 is intended to represent misalignment angles greater than 60° which, though rare, have occurred within the forty year history of the buoy data.

Changes to Table 1 are highlighted in yellow:

|  | Wind Speed (m/s) | Wind Direction (degrees) | Sign. Wave Height (m) | Wave Direction (degrees) | Wind-Wave Misalign. (degrees) |
|---|---|---|---|---|---|
| Metocean Case 1 | 41.7 | 0 | 12.4 | 0 | 0 |
| Metocean Case 2 | 41.7 | 0 | 10.0 | 30 | 30 |
| Metocean Case 3 | 41.7 | 0 | 11.0 | 60 | 60 |
| Metocean Case 4 | 34.9 | 90 | 12.4 | 120 | 30 |
| Metocean Case 5 | 38.6 | 60 | 12.4 | 120 | 60 |
| Metocean Case 6 | 41.7 | 0 | 12.4 | 120 | 120 |

Deletion in the first sentence in section 5.3 is shown below in red:

**5.3 Wind-wave misalignment**

Metocean case 1 considered an  approach whereby the maximum 50 year wind and wave conditions are applied without misalignment. The remaining cases considered varying amounts of wind-wave misalignment, as shown in Table 1. To investigate the impact of wind-wave misalignment on the jacket pile design, the differences for pile tension and compression forces are presented in Tables 14 and 15, respectively.

Changes to the 3rd paragraph in section 6 (Conclusions) are highlighted in yellow:

Finally, it was found that metocean case 1, which considered the maximum directional wind and wave values but did not account for wind-wave misalignment, did not consistently result in the highest axial pile loads. The loads from this case were underestimated by up to 6.9\% compared to cases including wind-wave misalignment. The cause of this difference could be traced back to the behavior of the turbine subjected to wind loading under various yaw errors. Caution must be exercised when considering whether or not to include wind-wave misalignment in a reduced load case table, for example during design stages prior to detailed design.

*#2: A brief discussion on the relative impact on the current loads to the overall total system loading should be included. The 2 m/s current load in line with the waves may overshadow the effect of the the wind loading on the structure. The likelihood of such an extreme current occurring simultaneously with the 50-yr wind and 50-yr wave for a given directional bin has a real recurrence interval of much larger than 50-yrs, and is likely overly conservative. The reviewer suggests running a sensitivity study of at least one additional current speed, 1 m/s for example, to determine whether the trends in wave directionality controlling optimum platform orientation hold constant.*

**Reply:** We appreciate this suggestion and decided to rerun the simulations considering a current speed of 1 m/s, as suggested. The findings from these additional simulations are incorporated in the revised article under a new section.

**Revision to article:** Section 5.4 has been introduced which states the following (highlighting omitted here for readability, but included in the separate track changes pdf):
* * *
**5.4 Current Speed**

In order to investigate whether the conservative current speed (2 m/s) might have influenced the findings, the simulations were rerun considering a reduced current speed of 1 m/s. As is to be expected, the pile top axial loads were overall reduced due to this change. Nevertheless, the trend elaborated in section 5.1, which demonstrates that the jacket orientation should be set relative to the wave direction, for the reference site conditions, remains valid for the reduced current speed (Figure 10). This figure also demonstrates that the axial pile top loads are not particularly sensitive to reasonable variations in the current speed.
* * *
[Figure]
* * *
**Figure 10.** Maximum pile top loads per jacket orientation (relative to the wave direction) for two current speeds
* * *
*#3: A consideration of DLC 1.6, where the turbine is operational with conditional 50-yr waves is considered in a full jacket design, and may be found to control the maximum member forces in certain members, especially near the top of the structure. At a minimum the wave conditions for DLC 1.6 near the rated turbine wind speed should be presented, along with a comparison of the turbine thrust loads at rated vs. idling with and without yaw error. With those data inference may be made as to whether DLC 1.6 might control for certain members and orientations.*

**Reply:** Additional simulations were performed to determine the pile top axial loads for DLC 1.6. The omni-directional conditional 50-year wave ($H_{s,SSS}$) was determined to be 9.5m. Buoy data entries with wind speeds above the turbine cut-out wind speed, after adjustment for the height difference, were filtered out and the Gringorten method was applied on the remaining dataset to determine this value. OpenFAST simulations were then run conservatively combining this conditional 50 year wave with the turbine's rated wind speed (10.59 m/s). Wind-wave misalignments of 0°, 30°, 60°, and 120° and jacket

orientations of 0°, 30°, and 60° were considered. A partial safety factor of 1.35 was applied and the results were seed averaged over six wind and wave seeds. The resulting pile top loads for DLC 1.6 and a comparison with the results presented in the article (DLCs 6.1 and 6.2) is provided in the figure below:

| Wind-wave misalignment | Jacket Orientation | Tension (MN) | | Compression (MN) | |
|---|---|---|---|---|---|
| | | DLC 6.x | DLC 1.6 | DLC 6.x | DLC 1.6 |
| 0° | 0° | 39.2 | 27.1 | 50.6 | 40.2 |
| | 30° | 31.8 | 20.8 | 61.9 | 53.5 |
| | 60° | 18.1 | 6.6 | 66.2 | 59.4 |
| 30° | 0° | 38.4 | 25.5 | 49.0 | 43.3 |
| | 30° | 30.9 | 22.7 | 63.9 | 48.5 |
| | 60° | 19.0 | 11.6 | 67.5 | 57.5 |
| 60° | 0° | 38.3 | 21.2 | 47.1 | 45.6 |
| | 30° | 29.8 | 20.8 | 63.9 | 42.1 |
| | 60° | 16.8 | 13.0 | 69.0 | 52.1 |
| 120° | 0° | 40.9 | 7.6 | 49.0 | 40.4 |
| | 30° | 33.5 | 9.5 | 62.5 | 35.1 |
| | 60° | 17.0 | 6.2 | 70.9 | 38.6 |

As can be seen in the figure, the pile top axial loads under DLCs 6.1 and 6.2 are higher than the loads for DLC 1.6 in all the considered cases. These findings support the statement made in the article under section 4.1: "These DLCs [6.1 and 6.2] were selected as the most likely governing load cases for the jacket pile embedment depth." Therefore, no changes are introduced in the article as a result of these simulations. Nevertheless, we hope this adequately addresses your comment.

*#4: A robustness check is often required for jackets, for example in API RP2A, which requires checking the jacket for 500-yr conditions without load factors, and can sometimes control the sizing and orientation of the jacket. How might the loads change if these environemntal conditions were to be introduced into the directional analysis?*

**Reply:** The 500-year directional extreme wind speeds were estimated using the available buoy data and reported below; including also a comparison to the 50-year values.

| | Extreme wind speed | |
|---|---|---|
| Direction | 50-year | 500-year |
| 0° direction | 41.7 m/s | 50.1 m/s |
| 60° direction | 38.6 m/s | 46.4 m/s |
| 90° direction | 34.9 m/s | 41.7 m/s |

Unfortunately, the 50-year extreme directional wave conditions are the maximum that can be simulated for the given jacket structure, due to the OpenFAST limitation described in paragraph two of section 3.2 in the article. The OpenFAST simulations were rerun considering a combination of the 500-year wind speeds, 50-year wave conditions, and a current speed of 2 m/s. Results of these rerun simulations and a comparison with the results presented in the article (using 50-year wind speeds) are presented in the figure below:

[Figure]

As is to be expected, the characteristic load level (not shown in the figure) increased due to these higher wind speed conditions. However, since the load factor for the robustness check is only 1.0 (compared to 1.35 and 1.1 for DLC6.1 and DLC6.2, respectively) the factored pile top axial loads (shown in the figure) decreased for some orientations. Interestingly, the optimal jacket orientation trend can still be seen even for these higher wind speeds, though the difference between orientations becomes less pronounced. It is expected that the difference would become more pronounced if the 500-year wave conditions had been combined with the 500-year wind.

The pile embedment depths presented in Table 11 of the article were also rechecked. The loads from this robustness check were found not to be governing for the pile embedment depths, since the material factor is also reduced to 1.0. It is possible that the pile embedment depths might have been governed by the robustness check if the 500-year wave conditions could have been considered. Though this is uncertain without performing the necessary simulations. Nevertheless, the figure above demonstrates that the optimal orientation should still be set relative to the wave direction even when considering 500-year environmental conditions, for the reference site conditions considered.

**Additional textual modification**

In the process of considering your feedback, a typo in Table 8 (which was Table 10 in the preprint) was noticed and has been corrected. Changes to Table 8 are highlighted in yellow:

|  | Description | Max Tension | Max Compression |
|---|---|---|---|
| Metocean Cases 2 & 3 | Highest wind speed | 30.2 MN | 61.3 MN |
| Metocean Cases 4 & 5 | Largest waves | 38.4 MN | 69.0 MN |
|  | **Percent Difference** | 23.9% | 11.8% |

The text describing this table, in the 1st paragraph of section 5.1, was correct in the preprint and remains unchanged in this updated revision.

Finally, we would like to thank both reviewers for your valuable comments and suggestions. We are convinced that your feedback has directly contributed to an improved update of the work, for which we are grateful. The following addition has been included in the updated revision of the article (highlighting omitted here for readability, but included in the separate track changes pdf):

> *Acknowledgements:* The authors wish to acknowledge the valuable feedback received from Thanh-Tuan Tran and an anonymous referee during the peer review process. Their comments directly contributed to the improvements implemented in the revision of this work, for which the authors are grateful. The authors also wish to acknowledge Alain Burgraeve (DEME Offshore) for his technical advice on the pile embedment calculations.

It is our hope that your comments have been sufficiently clarified in this reply and accordingly captured in the updated revision of the article. Thank you again for your contribution.